# Severe Acute Respiratory Syndrome Coronavirus 2 Variants of Concern: A Perspective for Emerging More Transmissible and Vaccine-Resistant Strains

**DOI:** 10.3390/v14040827

**Published:** 2022-04-16

**Authors:** Anacleto Silva de Souza, Vitor Martins de Freitas Amorim, Gabriela D. A. Guardia, Filipe F. dos Santos, Henning Ulrich, Pedro A. F. Galante, Robson Francisco de Souza, Cristiane Rodrigues Guzzo

**Affiliations:** 1Department of Microbiology, Institute of Biomedical Sciences, University of São Paulo, São Paulo 05508-000, Brazil; anacletosilvadesouza@usp.br (A.S.d.S.); vitormartins@usp.br (V.M.d.F.A.); rfsouza@usp.br (R.F.d.S.); 2Centro de Oncologia Molecular, Hospital Sírio Libanes, São Paulo 01308-060, Brazil; gguardia@mochsl.org.br (G.D.A.G.); fferreira@mochsl.org.br (F.F.d.S.); pgalante@mochsl.org.br (P.A.F.G.); 3Department of Biochemistry, Institute of Chemistry, University of São Paulo, São Paulo 05508-000, Brazil; henning@iq.usp.br

**Keywords:** SARS-CoV-2 variants, transmissibility, viral load, sensitivity to antisera

## Abstract

Novel severe acute respiratory syndrome coronavirus 2 (SARS-CoV-2) variants of concern (VOC) are constantly threatening global public health. With no end date, the pandemic persists with the emergence of novel variants that threaten the effectiveness of diagnostic tests and vaccines. Mutations in the Spike surface protein of the virus are regularly observed in the new variants, potentializing the emergence of novel viruses with different tropism from the current ones, which may change the severity and symptoms of the disease. Growing evidence has shown that mutations are being selected in favor of variants that are more capable of evading the action of neutralizing antibodies. In this context, the most important factor guiding the evolution of SARS-CoV-2 is its interaction with the host’s immune system. Thus, as current vaccines cannot block the transmission of the virus, measures complementary to vaccination, such as the use of masks, hand hygiene, and keeping environments ventilated remain essential to delay the emergence of new variants. Importantly, in addition to the involvement of the immune system in the evolution of the virus, we highlight several chemical parameters that influence the molecular interactions between viruses and host cells during invasion and are also critical tools making novel variants more transmissible. In this review, we dissect the impacts of the Spike mutations on biological parameters such as (1) the increase in Spike binding affinity to hACE2; (2) bound time for the receptor to be cleaved by the proteases; (3) how mutations associate with the increase in RBD up-conformation state in the Spike ectodomain; (4) expansion of uncleaved Spike protein in the virion particles; (5) increment in Spike concentration per virion particles; and (6) evasion of the immune system. These factors play key roles in the fast spreading of SARS-CoV-2 variants of concern, including the Omicron.

## 1. Introduction

The severe acute respiratory syndrome coronavirus 2 (SARS-CoV-2) rapidly became a concern due to its fast spreading, causing more than 390 million cases and 5.7 million deaths to date [1]. The SARS-CoV-2 genome has approximately 29,900 base pairs that encode four structural proteins, i.e., Spike protein (or *S. protein*), nucleocapsid (N), membrane (M) and envelope (E), in addition to 16 other non-structural proteins [2] (Figure 1a). In particular, the full Spike sequence has 1273 amino acids, having multiple functional domains distributed in two subunits, S1 and S2 [2] (Figure 1b). The S1 subunit recognizes the human receptor, allowing the virus to initiate its entry into the host cell [3], while the S2 subunit favors fusion of the virus envelope with the host cell membrane, thus facilitating virus entry into the host cell [3]. The S1 is composed of a N-terminal domain (NTD), a receptor binding domain (RBD), and SD1 and SD2 domains, while the S2 subunit is composed by the fusion peptide (FP), the heptapeptide repeat sequences 1 (HR1) and 2 (HR2), a transmembrane domain (TM), and an intravirion, palmitoylated cysteine-rich C-terminal domain, often referred to as the cytoplasmic domain or tail (CT) [3,4,5,6,7].

The functional role of S1 is associated with the NTD, which helps the virus to adapt to host or environmental conditions. The RBD recognizes and binds to human angiotensin-converting enzyme 2 (hACE2) [3]. The functional role of S2 is associated with the FP, responsible for the insertion of S2 into the target cell’s membranes [3,8]. The HR1 and HR2 form a bundle of helices responsible for the membrane fusion of the virus envelope and the host cell’s membrane into close proximity [9]. The TM is important for Spike protein trimerization and membrane fusion. The CT anchors the trimer in the viral membrane, and it is also involved in membrane fusion [8] (Figure 1).

**Figure 1 viruses-14-00827-f001:**
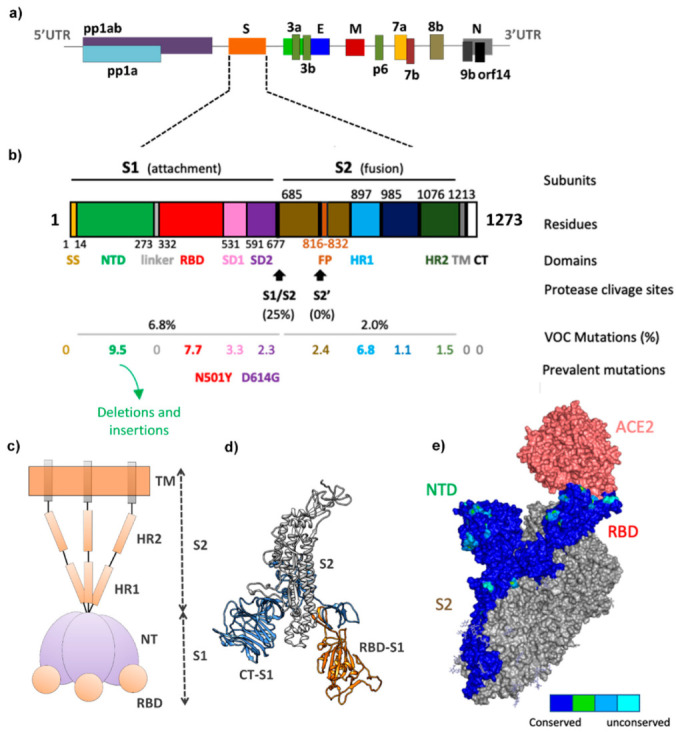
Structural features of SARS-CoV-2 Spike protein. (**a**) Genomic organization of SARS-CoV-2 based on the sequence of locus NC_045512.2 in the NCBI. (**b**) Domains in the Spike sequence. The SARS-CoV-2 Spike protein has two main subunits: S1 and S2, the first is related to the binding to the host cellular receptor, whereas the last allows the fusion of the viral and cellular membranes. During the infection process, the Spike protein is processed by a human serine protease, TMPRSS2, at the S1/S2 and the S2’sites to activate virus entry [10,11,12,13]. The total length of SARS-CoV-2 Spike is ~1273 amino acids and consists of a signal peptide (SP) located at the N-terminus followed by the S1 subunit composed of a N-terminal domain (NTD), a receptor binding domain (RBD) and SD1 and SD2 domains. The S2 subunit is composed of the fusion peptide (FP), the heptapeptide repeat sequences 1 (HR1) and 2 (HR2), a transmembrane domain (TM) and a cytoplasm domain (CT) [5,6]. The percentage of mutations in each domain of the Spike protein in relation to the Alpha, Beta, Gamma, Delta, and Omicron variants is shown. There are more mutations in the S1 (6.8%) than in the S2 domain (2%). Deletions and insertions have been observed in the NTD. Significant mutations have accumulated in the interface of interaction between the RBD Spike protein and ACE2. The S1/S2 protease cleavage site has 25% mutations, whilst no mutation was observed in the S2’ site. (**c**) Structural representation of the trimeric form of the Spike protein. (**d**) Monomeric Spike protein structure. (**e**) Conservation profile of the Spike protein among VOCs (Alpha, Beta, Gamma, Delta and Omicron) was performed using ClustalW server [14] followed by the ConSurf Server [15,16] (Appendix A). Surface of absolutely conserved residues are shown in only one subunit of the trimeric form of the Spike protein in dark blue and those less conserved in cyan, blue or green as shown in the bar (PDB ID 7DF4 [17]). ACE2 structure surface is colored in salmon, while the chains A and C in the trimeric form of the Spike protein are colored in grey.

Naturally, adaptive mutations occur in the S protein that interact with hACE2, mediating the viral infection of the host cell [18,19]. Mutations in the Spike protein may improve or worsen infectivity, transmissibility, and pathogenicity, compared to the wild-type SARS-CoV-2 (SARS-CoV-2^WT^) [20]. In 2021, the main variants, denominated Alfa (B.1.1.7), Beta (B.1.351), Gamma (P.1) and Delta (B.1.617.2), spread worldwide, bringing more uncertainties about the pandemic course [21]. The Delta variant became the dominant variant (from 10% to over 90%) in less than 2 months in the studied countries (Appendix A) [22]. In November 2021, a novel variant denominated Omicron (B.1.1.529) was reported, presenting a much higher number of mutations (26 to 32 mutations only in the Spike protein) and raising concerns regarding how it would behave [23]. To date, the world is facing a new wave due to SARS-CoV-2^Omicron^, which is highly infectious and is becoming the dominant variant worldwide.

Turkahia and collaborators showed evidence of recombination, mainly between the SARS-CoV-2^Alpha^ and SARS-CoV-2^B.1.177^ variants and found that most recombination breakpoints were located within the gene coding for the Spike protein [24]. Another issue is the observed cases of co-infection by SARS-CoV-2 and Influenza [25,26]. RNA recombination between coronaviruses and Influenza C has been reported [27], suggesting that a novel SARS-CoV-2 variant that contains genetic information from both viruses might emerge. Such possibilities make the scientific and medical communities worry about vaccine effectiveness, symptoms, disease severity, transmissibility and even the efficiency of current diagnostic tests for SARS-CoV-2.

Molecular dynamics simulations and cryo-electron microscopy have shown that the dominant substitution D614G favors the up-conformation of the RDB of the Spike trimer [28,29,30], which is more accessible for interacting with hACE2 [3,6,31,32]. In the case of SARS-CoV-2^WT^, only one out of three RBDs is in the up-conformation. As for SARS-CoV-2^Delta^, all three RBDs are in the up-conformation state, increasing the likelihood of hACE2 interaction (Appendix A). Mutations, especially in the RBD, increase the virus capability to infect and to become more transmissible and resistant to the immune system [33,34,35,36,37]. Furthermore, specific mutations may alter the equilibrium dissociation constant (*K*_D_), the association constant (*k*_a_) and the dissociation constant (*k*_d_), all critical properties for the interaction with hACE2 [38,39]. Experimental studies demonstrated that SARS-CoV-2 is evolving to resist neutralizing antibodies. It has been shown that the E484K mutation in the RBD enables viruses to evade neutralizing antibodies and this residue is one of the most variable residues among the VOCs [40] (Appendix A). This mutation is localized in the main antibody-recognizing region [41]. The Delta variant has been described as more transmissible than the Alpha variant [42] and presents less susceptibility to neutralizing antibodies mobilized by vaccines than other SARS-CoV-2 VOCs do [43]. Computational studies have shown that such capabilities may be caused by structural modifications because of L452R and T478K mutations in the RBD [44,45]. Interestingly, T478 is also a highly variable residue among the VOCs (Appendix A).

Several vaccination technologies have been developed and are being used to immunize the population worldwide. Vaccination was shown to be effective in significantly reducing the death rate for patients with COVID-19, yet not blocking the transmission of the virus, which still occurs in an apparently reduced way [46]. However, the investigation of underlying mechanisms behind emerging SARS-CoV-2 variants resistant to neutralizing antibodies induced by vaccines or previous virus infections has gained prominence in the scientific community [47]. In the present review, we approached the impacts of Spike protein mutations on infectivity and transmissibility of SARS-CoV-2 VOCs. Additionally, we discuss the capacity of the VOCs to evade neutralizing antibodies or decrease the sensitivity to antisera from convalescent and vaccinated patients, appearing to be the main factor driving the evolution of the virus [48]. We compile evidence from different studies that show the Spike protein in the SARS-CoV-2 VOCs evolved in distinct ways to present an increase in infectivity in the host cell, with higher infection fitness than the wild type. Since emergence of higher fitness VOCs is expected while opportunities for mutation and recombination continue to be provided by high levels of virus circulation in the host population, we expect continued immunization and restrictive measures to continue to be the main strategies to diminish the likelihood of VOC emergence until effective drugs that block the virus infection are made available.

## 2. SARS-CoV-2 VOC Mutations and Their Impact on hACE2 Binding Affinity and Escape from the Immune System

The evolution of the SARS-CoV-2 Spike protein has led to key structural features that facilitate fast-spreading transmission. The D614G mutation in the Spike protein was the first that worried the scientific community during the coronavirus disease 2019 (COVID-19) pandemic and is still present in all SARS-CoV-2 VOCs. This SARS-CoV-2 mutation affects viral replication in lung cells and viral infectivity. The D614G substitution does not alter Spike biosynthesis, processing, or cell–cell fusion of SARS-CoV-2, but it disrupts an interprotomer contact breaking a salt bridge between D614 and K854 located at the FPPR (fusion peptide proximal region). Thus, favoring an RBD up-conformation state to bind to hACE2 but does not increase protein stability [32,49]. Molecular dynamics simulations showed that conformational changes in the D614G Spike are energetically favorable for increasing infectivity due to an enhanced exposure of the RBD to interact with hACE2 [28]. This favorable structural conformation may be explained due to a displacement of the loop composed of residues 620 to 640 [49]. This substitution also promotes an increase in the number of functional Spikes [50]. Furthermore, the D614G mutation moderately increased binding affinity to hACE2, providing a clear selective advantage to these mutants compared with ancestral SARS-CoV-2 (Table 1) [49].

The main SARS-CoV-2 VOCs are Alpha, Beta, Gamma, Delta and Omicron, in which the main mutations are localized in the Spike protein. In the Spike S1 subunit, the main changes happen in the NTD and RBD domains, while in the S2 subunit it happens in the HR1 domain (Figure 1b). Analysis of residue variability along the VOCs shows that Y145, K417, T478, E484 and H655 are very variable and most of them are exposed or involved in the hACE interaction (Figure 1e and Appendix A). This suggests that they may be important for evading the immune system or for modulating human receptor binding. The NTD and RBD are the main targets of neutralizing antibodies and the elevated number of mutations observed in these domains for novel SARS-CoV-2 VOCs suggests that the virus might be evolving to escape the neutralizing action of antibodies produced by immunization or earlier infections. Appendix A shows the multiple sequence alignment of the Spike protein variants from SARS-CoV-2 VOCs. The Alpha variant, first detected in the United Kingdom, has mutations Δ69–70, Δ144, N501Y, A570D, D614G, P681H, T716I, S982A and D1118H. The Beta (B.1.351) variant, first reported in South Africa, presents L18F, D80A, D215G, Δ242–244, R246I, K417N, N501Y, D614G and A701V mutations. The Gamma (P.1) variant, first identified in Brazil, presents mutations L18F, T20N, P26S, D138Y, R190S, K417T, E484K, N501Y, D614G, H655Y, T1027I and V1176F. The Delta (B.1.617.2) variant, first identified in India, has mutations T19R, Δ157–158, L452R, T478K, D614G, P681R and D950N in the Spike protein. Finally, the Omicron (B.1.1.529) variant, first reported also in South Africa, has mutations in A67V, Δ69–70, T95I, Δ142–144, Y145D, Δ211, L212I, insertion of EPE at position 214, G339D, S371L, S373P, S375F, K417N, N440K, G446S, S477N, T478K, E484A, Q493R, G496S, Q498R, N501Y, Y505H, T547K, D614G, H655Y, N679K, P681H, N764K, D796Y, N856K, Q954H, N969K and L981F [63].

Mutations in the Spike protein correlate to enhanced virus fitness (such as the prevalent mutations, N501Y and D614G), increased binding to the hACE2 (such as N501Y) and resistance against neutralizing antibodies (highly variable residues among VOCs, such as T478 and E484). All these mutations cause conformational changes in the Spike trimer structure. Indeed, cryo-EM structures of the Alpha variant spike protein reveal a rotation of all three S1 subunits leading to up-conformation of the RBD [61]. When compared with D614G, the SARS-CoV-2^Beta^ Spike presents a similar conformational state, differing only in the NTD of the Spike S1 subunit [61]. The triple-residue deletions (L242, L243 and A244) in the SARS-CoV-2^Beta^ Spike protein result in structural changes in the adjacent loop (residues 246 to 260) and the nearby loop (residues 144 to 155), both of which form part of neutralizing epitopes [61]. These structural changes may affect the binding of neutralizing antibodies [61] and likely affect the efficiency of currently used COVID-19 vaccines. The cryo-EM structure reveals that RBDs of SARS-CoV-2^Gamma^ Spike protein mainly visit the up conformation [64]. The change in L452R in the Spike protein of SARS-CoV-2^Delta^ may contribute to 50% more transmissibility than the Alpha variant [65]. It could be correlated with electrostatic interactions with hACE2 [44,45].

Each SARS-CoV-2 VOC presents different values of *k*_a_, *k*_d_ and *K*_D_ for the interaction between the RBD or the trimeric form of the Spike protein and hACE2 (Table 1). Veesler and co-authors performed binding affinity assays to hACE2, comparing the trimeric Spike from SARS-CoV-2 variant with SARS-CoV-2^WT^, showing a ratio *K*_D_^VARIANT^/K_D_^WT^ of ~1:5 [6]. When compared to SARS-CoV, SARS-CoV-2 Spike *k*_a_ kept the same value of ~1.4 × 10^5^ M^−1^ s^−1^ but decreased its *k*_d_ value from 3.0 × 10^−4^ s^−1^ to ~1.7 × 10^−4^ s^−1^. In relation to the Spike of SARS-CoV-2^WT^, D614G Spike trimer presents comparable *k*_a_ and *k*_d_ rates of ~1.6 × 10^5^ M^−1^ s^−1^ and ~1.7 × 10^−4^ s^−1^, resulting in similar binding affinity to hACE2 (*K*_D_^SARS-CoV−2^/*K*_D_^D614G^ of ~1:1) [60].

Our previous studies have shown that the Spike protein of SARS-CoV-2^Alpha^, SARS-CoV-2^Beta^ and SARS-CoV-2^Gamma^ presents comparable values of binding affinity to hACE as SARS-CoV-2^WT^, with *K*_D_ values ranging from 1.1 to 1.8 nM [54], but higher affinity than the trimeric form of the Spike protein of SARS-CoV-2^WT^ (*K*_D_ values ~16 nM) [55]. A similar result was also observed for the Omicron variant that has higher affinity to hACE than the wild type but comparable values of binding affinity to hACE as SARS-CoV-2^Beta^ [66]. The main significant differences seem to be in the *k*_a_ and *k*_d_ values, ranging 0.1–0.3 × 10^5^ M^−1^ s^−1^ and 1.7–3.0 × 10^−4^ s^−1^, which are less than those of SARS-CoV-2^WT^ (*k*_a_ and *k*_d_ values of 0.4 × 10^5^ M^−1^ s^−1^ and 7.0 × 10^−4^ s^−1^, respectively) [55]. Lower differences in the *K*_D_ values, but larger ones in *k*_a_ and *k*_d_ values, imply changes to the binding kinetics to hACE2. Decreasing *k*_d_ values due to mutations in these SARS-CoV-2 variants increase the likelihood of the Spike protein being cleaved by proteases, vital to membrane fusion and virus cell host entry. Therefore, these variants seem to improve virus entry in the host cell by increasing the probability of cleavage of the Spike protein by proteases, a step that is a requirement for the process of membrane fusion [55].

The emergence of novel SARS-CoV-2 VOCs with significant mutations in the Spike protein might result in changes in the virus tropism. This can occur when the proteases cleavage sites are mutated, as already observed for the Alpha, Delta and Omicron variants (Appendix A), or if the Spike protein starts to recognize other human receptors, expanding the repertoire of entry points of the virus to new human cell lineages. This would result in different clinical symptoms of the disease and a significant decrease in efficiency of previous immune responses, acquired by previous infections or immunization based on SARS-CoV-2^WT^. This observation has been reported for the Omicron variant, which replicates faster in the airways and has an increased fitness compared to the D614G and Delta variants [67]. The replication of the Omicron variant in alveolar type 2 cells is not productive and does not efficiently use TMPRSS2 for entry or spread through cell–cell fusion [67]. Omicron has an altered protease usage and tropism, as shown in animal model studies, features that are probably related to this variant’s decreased pathogenicity compared to previous variants [67].

## 3. SARS-CoV-2 VOC Decreases Incubation Period, Increasing Viral Loads, Transmission Period and Transmissivity

One of the factors that contributed to SARS-CoV-2 becoming a pandemic virus is that viral transmission occurs in asymptomatic infected individuals and before the appearance of symptoms in symptomatic cases. The kinetics of the SARS-CoV-2 infection involve contact with the virus, viral incubation, period of viral transmission (viral shedding) that is associated with increased viral load in the infected person, the period of symptom onset and detection of viral RNA (viral RNA shedding) by diagnostic tests (Figure 2) [68,69]. The degree of viral transmissibility can be measured by the number of people who are infected for each person previously infected when everyone is susceptible. This number is called effective reproduction number R_0_ [70]. In general, the highest viral loads are reached when symptoms appear, gradually declining until vanishing around 21 days after the onset of symptoms. However, the live virus is only detectable up to the eighth day after the onset of symptoms [71], which may decrease transmissibility after this period. Viral loads are similar across age, sex and disease severity [72]. Interestingly, men transmit SARS-CoV-2 more effectively than women do [73]. This may be explained by the observation that men, aged 48 or less, have a viral load about ten times higher in the saliva than women [73]. Nevertheless, no differences were observed, in this cohort, for samples of nasofaringe, thus suggesting such biases may not apply to variants with different tropism.

Considering the SARS-CoV epidemic context, estimated *R*_0_ values ranged from ~0.5 to 1.3 [74]. Conversely, COVID-19 pandemic showed a compelling increase in *R*_0_ values, ranging from 1.4 to 3.9 [75] (Table 2). In general, SARS-CoV-2 VOCs tend to have higher *R*_0_ values relative to the wild-type SARS-CoV-2. In particular, the substitution D614G in the Spike protein makes SARS-CoV-2 31% more transmissible (*R*_0_ value ranging from 1.7 to 4.7) [76]. Shi and co-authors showed that hamsters infected with SARS-CoV-2 expressing D614G Spike mutant may also reveal increased virus transmission [77]. Increases in severity and mortality of COVID-19 are not associated with this D614G mutation [20], since the Alpha, Beta and Delta variants all share this substitution. The higher transmissibility associated with this mutation seems to derive from an increase in viral load in younger patients [20]. The Alpha variant, whose estimated *R*_0_ interval is from 2.2 to 6.1 (Table 2), is from 43 to 90% more transmissible than SARS-CoV-2^WT^ [78]. The Beta variant is ~50% more transmissible than the SARS-CoV-2^WT^ [79] and it is estimated to have an *R*_0_ ranging from 2.1 to 5.5 (Table 2). The Gamma variant is ~40–120% more transmissible than SARS-CoV-2^WT^ [80], which corresponds to an estimated *R*_0_ ranging from 2.1 to 5.5 (Table 2). The Gamma variant evades neutralizing antibodies, leading to possibly higher rates of SARS-CoV-2 reinfection [81], and the same might happen to the other SARS-CoV-2 VOCs. The *R*_0_ of the Omicron variant is estimated to be as high as 10. As a result, the control of viral transmission in the UK is almost impossible since the cases of Omicron are doubling every 2–3 days [82], testing has been suggested to help to control the virus spreading [82].

SARS-CoV-2 VOCs modulate the kinetics of the SARS-CoV-2 infection in a way that improves the virus fitness in the human body (Figure 2), thus leading to reduced vaccine effectiveness and, in some cases, to increased disease severity. Concerning the Delta variant, which became the dominant lineage worldwide in 2021, the transmissibility rate (R_0_) increased from 1–4 to 6.4 [70,83], driven by higher viral loads (PCR Ct values) [68,84], shorter time to peak viral load (shorter incubation period [70,83]), longer viral shedding (slower decline) and abrogated neutralization capacity compared to non-Delta SARS-CoV-2 variants [43] (Table 2). Interestingly, the Delta variant is associated with higher odds of oxygen requirement, intensive care unit admission or death [68]. One of the explanations for the fast spreading of the Delta variant is its higher viral replication rate (2.7) [85] (Table 2), which results in higher viral loads and shorter incubation time. Nonetheless, regarding the Delta variant, the symptoms onset is about 5.8 days after infection, not very different from wild-type SARS-CoV-2 [72,86] (Table 2). Consequently, there is an increasing period of time in which infected people are spreading the virus in the presymptomatic stage. Subjects infected with the Delta variant may commence transmission 1.8 days prior to symptoms onset, compared to 0.8 days for previously SARS-CoV-2 variants. Consequently, it was estimated that 44% of the secondary cases were infected by presymptomatic people, and this number may have raised up to 74% for the Delta variant [72,86]. Since December of 2021, the Omicron variant was found to spread even faster than the Delta variant mainly due to its ability to escape the immune response, even as this variant is seemingly less lethal than previous ones. The Omicron variant has become the most prevalent variant in the beginning of 2022 worldwide.

**Table 2 viruses-14-00827-t002:** Biological features of SARS-CoV, SARS-CoV-2, and SARS-CoV-2 variants. The median incubation period for the Alpha variant has been estimated at around 3 days, compared to around 5 days for ancestral strains [87,88,89]. The Delta variant has a shorter incubation period when compared to ancestral strains (4 days vs. 6 days) [89,90]. Additionally, increased S-protein density in the virion as well as the increase in the S1/S2 ratio correlate with this increased infectivity due to mutation D614G. D614G increases the stability of the Spike protein, decreasing the natural loss of S1 subunit, thus enhancing functional Spikes into the virion. D614G does not affect affinity to hACE2 [50,91].

Strains	Spike Density (Unit/Virion)	Mean S1/S2 Ratio &	Viral Load (Mean C_T_) #	Initial Viral Load (Mean C_T_) *	Days of Virus Incubation	Days of Viral Shedding **	R_0_	Growth Rate, log_10_ Units Per Day
SARS-CoV	50–100 [92]	1.1 [93]	26.9 [94]	ND	ND	ND	0.54–1.3 [74]	ND
SARS-CoV-2^WT^	11–41 [95]	1.0 [91]	21.2 [96]	28 [68]	5–7 [90,97]	13 [90]	1.4–3.9 [75]	3.2 [85]
SARS-CoV-2^D614G^	28–103 [97]	1.0 [91]	19.9 [96]	ND	ND	13 [90]	1.7–4.7 [76]	ND
SARS-CoV-2^Alpha^	28–103 [97]	1.2 [91]	17.4 [98]	22 [68]	3 [90]	13 [90]	2.2–6.1 [78]	3.1 [85]
SARS-CoV-2^Beta^	28–103 [97]	1.2 [91]	18.9 [99]	22 [68]	ND	13 [90]	2.1–5.5 [42]	ND
SARS-CoV-2^Gamma^	28–103 [97]	1.2 [91]	19.8 [100]	ND	ND	ND	4.7–4.9 [80]	ND
SARS-CoV-2^Delta^	ND	ND	ND	18 [68]	4–6 [86,90]	18 [90]	5.0–8.0 [89]	2.7 [85]

# Viral load is determined using RT-PCR from nasal swab samples. * PCR Serial cycle threshold values in the beginning of the COVID-19 symptoms. ** Mean number of days after COVID-19 initial symptoms for the patient to decrease the viral load (PCR C_T_ (mean) bigger than 30). ND = not determined.

## 4. SARS-CoV-2 VOCs Present Less Sensitivity to Neutralizing Antibodies Than Ancestral SARS-CoV-2

Currently, more than 180 anti-SARS-CoV-2 vaccine candidates are in clinical trials. These candidates include inactivated, live attenuated, recombinant protein, vectored, RNA-based and DNA-based vaccines [101].

*Inactivated vaccines*. Inactivated vaccines are developed by growing viruses in cell culture followed by chemical or heat inactivation. Inactivated vaccines include CoronaVac, which are administered intramuscularly with adjuvants [102]. In Brazil, CoronaVac has shown, after two doses, 51% efficacy against symptomatic cases and 100% against hospitalization and mortality caused by COVID-19 [103]. A phase 3 trial in Turkey has shown an efficacy of 86.3% against the Alpha variant and 96.4% against non-Alpha variants [104]. CoronaVac was shown to be tolerable and to present immunogenicity and safety in children and adolescents from 3 to 17 years and healthy adults aged from 18 to 59 years [105,106]. Covaxin (BBV152) is also an inactivated virus vaccine, which stimulates a protective immune response and, in phase 3 trial data, has an efficacy of 77.8% [107]. It was demonstrated that neutralization activities of sera from vaccinated people with BBV152 presented the same efficacy for the Alpha variant [108]. However, neutralization activity of sera collected from convalescent patients and vaccinated individuals with two doses of BBV152 demonstrated a decrease in neutralization titers against Beta and Delta [109]. Notwithstanding, these sera still showed a protective response against these variants [109]. BBV152 presented 65.2% protection against the Delta variant in Phase 3 clinical trial [107].

*Live attenuated vaccines and recombinant protein vaccines.* Live attenuated vaccines are produced by a genetically weakened virus version, which induces an immune response similar to that occurring upon natural infection. Recombinant protein vaccines involve the use of recombinant Spike protein. An example for that is Spike injection, such as in the case of the Novavax vaccine [110]. The Novavax (NVX-CoV2373 or Covovax) was developed by Novavax and the Coalition for Epidemic Preparedness Innovations (CEPI). In a randomized study, after two doses of the NVX-CoV2373, the vaccine protected 89.7% against the SARS-CoV-2 infection and showed 86.3% efficacy against the Alpha variant [110].

*Replication-incompetent vectors.* Replication-incompetent vectors are based on another virus that presents partial deletions of its genome and is able to express the Spike protein. These vaccines use vectors, for example, adenovirus, human parainfluenza virus and influenza virus. An adenovirus-based vaccine, Gam-COVID-Vac (Sputnik V), showed good safety and has shown to induce immune responses in participants aged 18 years or older [111]. The vaccine was administered in two doses intramuscularly for 21 days using two different recombinant adenovirus vectors (rAd26 and rAd5), both of them presenting the gene for the full-length SARS-CoV-2 Spike protein [111]. The Gam-COVID-Vac showed good efficacy (91.6%) and was tolerable in most participants [111]. Oxford-AstraZeneca vaccine (AZD1222) is recommended for people aged 18 years or older. A randomized study performed in Brazil, South Africa and the United Kingdom revealed an efficacy of 70.4% after participants received two doses (each dose containing 5 × 10^10^ viral particles) [112]. The vaccine of Janssen Pharmaceuticals Companies of Johnson & Johnson (JNJ-78436735) is also recommended for people of 18 years or older and had an efficacy of 63.3% in a clinical trial with people who had received only one dose [113].

*RNA-based vaccines.* RNA-based vaccines are recent developments and similar to DNA-based vaccines, use the genetic information to produce the antigen in the cells. Currently, mRNA-based vaccines produce recombinant Spike protein and are developed by Moderna and Pfizer-BioNTech. Moderna COVID-19 Vaccine (mRNA-1273) is recommended for people aged 18 years or older and presented an efficacy of 94.1% after two doses [114]. Specifically, vaccine effectiveness against the Alpha variant was 88.1 and 100% after the first and second doses, respectively [115]. However, the effectiveness against Beta was 61.3 and 96.4% after first and second doses, respectively [115]. The Pfizer-BioNTech vaccine (BNT162b2) is recommended for people 12 years or older [116]. In addition, the Pfizer-BioNTech vaccine is approved for individuals who are 5 years of age or older. However, in the present data, safety and effectiveness of this vaccine in children younger than 5 years have not yet been established [117]. Similarly, the Pfizer-BioNTech vaccine efficacy was 95%, protecting against SARS-CoV-2 in people who received two doses [116]. Notably, low efficacy was found after only 1 dose of BNT162b2 and AZD1222 vaccine, presenting ~34% against Delta and ~51% against Alpha [118]. After the second dose of BNT162b2, on the other hand, efficacy was 93.4% for Alpha and 87.9% for Delta variants. Likewise, AZD1222 presented an efficacy of 66.1% for Alpha and 59.8% for Delta variants [118].

*Virus is evolving to evade the immune system and to increase its infectivity*. The emergence of novel SARS-CoV-2 variants is expected, though the selection of variants that circumvent the neutralizing effect of plasma from convalescent and immunized patients is under concern [119]. The course of the virus evolution in order to survive involves evading the immune system and infecting more and more people, spreading further and increasing its population. This hypothesis has been related to the decreasing sensitivity to neutralizing antibodies from infected patients with SARS-CoV-2, exerting a selective pressure, with a major concern of emerging resistant SARS-CoV-2 variants [120].

Nussenzweig and co-authors showed that convalescent plasma samples had less than 50 titers in 33% of infected individuals, more than 1000 in 79%, and more than 5000 in 1%, demonstrating that most convalescent plasma samples did not have high levels of neutralizing antibodies [121]. The sensitivity of neutralizing antibodies produced by convalescent patients or induced by vaccines is higher in SARS-CoV-2^WT^ and, in general, the performance of this sensitivity is decreased against any SARS-CoV-2 VOCs (Table 3). According to Table 3 and Table 4, the Beta SARS-CoV-2 variant, in relation to the Alpha variant and the wild type, is more resistant to convalescent patient antisera and vaccine-induced antibodies. Indeed, sera from convalescent patients after 12 months of SARS-CoV-2 infection show that neutralizing antibodies is 4-fold less sensitive against Delta than Alpha [43]. Furthermore, only a single dose of vaccines, either AstraZeneca or Pfizer, did not protect against Delta infection, while two doses generated efficient immune responses against this variant [43]. These data suggest that SARS-CoV-2 variants that elicit a less effective immune response or that are more resistant against neutralizing antibodies are more successful and become the source of worldwide fast-spreading waves of virus infections, thus extending the duration of the pandemic.

When compared to sera from convalescent patients infected with SARS-CoV, antisera from SARS-CoV-2-infected convalescent patients presented a significant loss of sensitivity to the virus (ID_50_^SARS-CoV^ range from 1500 to 8000, while ID_50_^SARS-CoV−2^ has value of 1402) (Table 3). Evidently, the sensitivity to neutralizing antibodies obtained from convalescent patients and vaccinated patients (mRNA-1273, Moderna) showed similar profiles in the SARS-CoV-2^WT^ and D614G variants (Table 3 and Table 4). Sensibility reduction was observed in neutralization of Kappa (B.1.617.1) and Delta (B.1.617.2) by antisera obtained from convalescent patients and vaccinated individuals. A loss of protection efficacy of 3.9-fold for convalescent plasma, 2.7-fold for the Pfizer-BioNTech vaccine and 2.6-fold for the Oxford-AstraZeneca vaccine was observed for the Kappa variant. For infection by Delta, protection rates decline 2.7-, 2.5- and 4.3-fold, respectively. Such reduced efficacies were comparable in scale with those seen for Alpha and Gamma, with no evidence of widespread escape from neutralization, in contrast to that observed for Beta. These results make it likely that the current RNA and viral vector vaccines will provide protection against the B.1.617 lineage, although an increase in breakthrough infections may occur as a result of the reduced neutralizing capacity of sera. Unfortunately, after two doses of CoronaVac, neutralizing antibodies seem to be less effective against any SARS-CoV-2 variants, mainly the Delta variant (Table 4). In the case of the Omicron variant, convalescent patients or immunized patients with Ad26.COV2.S (single dose), BBIBP-CorV or Sputnik V had no neutralizing activity against Omicron with the exception for one Ad26.COV2.S- and three BBIBP-CorV-immunized patients’ serum. Serum of immunized individuals with mRNA1273, BNT162b2 and AZD1222 displayed higher neutralization against Wuhan-Hu-1 and activity against Omicron with a decrease of 33-, 44- and 36-fold, respectively. Interestingly, serum from vaccinated cohorts who were previously infected displayed higher neutralizing antibodies with a decrease of 5-fold [66]. Therefore, the SARS-CoV-2 population seems to be evolving towards strains that are less affected by neutralizing antibodies induced by vaccines or a previous SARS-CoV-2 infection.

It is worth mentioning that antibody-dependent enhancement (ADE) is an alternative mechanism that some viruses use to infect cells [129,130,131]. This happens when the virus binds to receptor molecules, known as Fcγ receptors (FcγRs), on immune cells such as macrophages or monocytes and uses these receptors as a route to internalize the virus. Antibodies to any viral epitope with low affinity or in sub-optimal titer can induce ADE [132]. In the case of infection by SARS-CoV-2, the antibodies produced may elicit ADE following infection [133,134] primarily by the interaction with two types of FcγRs, FcγRIIA and FcγRIIIA. Nevertheless, no virus replication in macrophage cells was observed [133,134,135]. This mechanism seems to not be correlated to aberrant cytokine release by macrophages during some cases of SARS-CoV-2 infection, still it may function as a mechanism to trap the virus in the macrophages [133].

## 5. SARS-CoV-2 VOC Spike Protein Evolved in Different Ways to Facilitate Virus Spreading and Evasion of Neutralizing Antibodies

The Spike surface protein of main SARS-CoV-2 VOCs have evolved to increase viral fitness and facilitate the virus spreading (Figure 2). For this, at least six different mechanisms have been described that increase the efficiency of infection by SARS-CoV-2 VOCs and are directly related to mutations in the Spike protein (Appendix A): (1) increasing hACE2 affinity (*K*_D_) and (2) extending the time the Spike protein remains bound to hACE2, thus increasing the likelihood that Spike is cleaved by proteases and proceeds to membrane fusion. (3) The Spike D614G mutation favors the up conformation of the RBD in the trimeric state of the protein, amplifying the amount of Spike protein subunits able to bind to hACE2. This mutation also (4) better stabilizes the trimeric form of the Spike protein (its uncleaved form). The Spike protein may spontaneously shed its S1 subunit, and this early cleavage leads to protein inactivation, preventing virus infection. The Spike D614G mutation boosts the amount of Spike proteins in the surface of the virion able to bind the human receptor and, as a result, increases the infectivity rate. (5) Mutations in L452R, T478K and E484K located at the RBD increase the resistance to neutralizing antibodies. Therefore, it is a mechanism to evade the immune response and an alert to possible new mutations in Spike that could impair the effectiveness of current vaccines. Finally, (6) the Spike D614G mutation also raises the number of Spike proteins per viral particle. This observation implies a shift in the chemical equilibrium Spike + hACE2 ⇌ Spike-hACE2 is expected and an increase in Spike protein numbers per viral particle should be enough to increase the effectiveness of viral infection, as it elevates the amount of Spike bound to hACE2. Interestingly, this phenomenon could cause the virus to bind to tissues with a low amount of hACE2, which could change the severity and pathology of the disease. It may also result in a decrease in the viral dose required to cause infection. Taken together, all mechanisms described above end up helping Spike’s access to its ACE2 receiver and increasing its infection success, resulting in more people infected in a shorter period of time. These mechanisms may also explain why the Spike protein of the Delta variant fused membranes more efficiently at low levels of the cellular receptor ACE2 [22].

There are five residues that are highly variable among SARS-CoV-2 VOCs: Y145, K417, T478, E484 and H655 (Appendix A), most of them are exposed in the Spike protein’s surface (Appendix A) and are related to evading the immune system. Interestingly, SARS-CoV-2^Omicron^ carries the mutations T478K, E484A and D614G, all of them important to evade the immune system and for the fast spreading of SARS-CoV-2. Moreover, mutations in the RBD of the Omicron variant, such as K417N and N501Y, are predominantly distributed in the interface of its interaction with hACE2 and may have synergistic actions for escaping from neutralizing antibodies [48,136,137]. In animal model studies, the Omicron variant was shown to be less pathogenic, i.e., to cause milder symptoms, while being 10–20% more transmissible than the Delta variant. Intriguingly, the Omicron variant also showed to outcompete the Delta variant under immune selection pressure [138]. In addition to the effect of the mutations described in the Spike protein other mutations found in other SARS-CoV-2 proteins may contribute in different ways to increase virus fitness, infectivity, transmission, severity of the COVID-19 infection (such as the P4715L in ORF1ab [139]) and may also facilitate human SARS-CoV-2 reinfection. Interestingly, some mutations in the structural protein nucleocapsid increase mRNA packaging and delivery, RNA content per virion, resulting in increased virus replication [140].

## 6. Epidemiology of COVID-19

According to the World Health Organization (WHO), an increase in the number of COVID-19 cases can be observed in the second half of 2021 in countries from the Northern Hemisphere (Figure 3). Even after the vaccination rates in the United States, United Kingdom, Germany and Japan surpassed 50%, all these countries had a significant increase in the number of cases and deaths. These data are explained by the emergence of novel variants of SARS-CoV-2 associated with lower efficiency of neutralizing antibodies [54,141]. For those countries that surpassed 60% of the population vaccinated with the first dose, such as Germany and the United Kingdom, the number of deaths was remarkably lower than rates seen during the first wave. Unlike these countries, the United States and Japan had 55.6% and 37.2% of the population vaccinated with the first dose, leading the number of deaths to exceed 50% compared to the first wave of the COVID-19 pandemic. Since Japan has very peculiar characteristics in its historical and behavioral context [142], death and case rates dropped in 11 weeks. However, in the period of the 2021 Olympic Games [143], the country had a significant increase in the number of COVID-19 cases, showing a direct relationship between collective activities, increased circulation and the time of transmission of the Delta variant [144] (Figure 3). It is very evident that the vaccine plays a key role in decreasing death rates from COVID-19. It is estimated that the rate of hospitalization among unvaccinated individuals is 29.2 times higher than those fully vaccinated [145]. Figure 3 shows the prevalence of the Delta and Omicron variants in the United Kingdom, Japan, United States and Germany and the timing of the increase in COVID-19 cases. Delta variant predominance increased from 10 to 90% in the range of 49 days in the USA, 52 days in Japan, 49 days in the UK and 40 days in Germany (Appendix A). In part, the flexibility of sanitary measures probably contributed to these trajectories, but the emergence of the Omicron variant at the end of 2021 was the main reason for the staggering increase in the rates of the COVID-19 infection at the beginning of 2022 (Figure 3). The UK eased the use of masks on 19 July 2021 [146]. On 16 April 2021, New Hampshire, United States, also relaxed its restrictive measures [147]. Therefore, even with high vaccination rates, the higher numbers of infected people may in part be explained by the relaxation of restrictive measures, which allowed the virus to spread more easily, increasing the likelihood for the emergence of novel variants able to take advantage of any of the several mechanisms, hereby discussed, to evade the immune response, replicate faster or to higher loads and/or to transmit more efficiently (Figure 4). 

## 7. Conclusions

In this review, we discussed the SARS-CoV-2 VOC mutations and their impacts on viral infectivity and transmissibility. SARS-CoV-2 has evolved, and novel VOCs are still emerging. The route of the actual pandemic is still unknown, but mass worldwide vaccination has been effective to reduce the risk of severe illness, hospitalization, and death from COVID-19, although vaccines were mostly unable to block the spread of the virus. In this regard, restriction measurements to reduce virus spread are also essential to reduce the chances of emergence of new variants capable of evading the immune system, spreading much faster, and with a different tropism than previous strains. All SARS-CoV-2 VOCs are of clinical concern, putting at risk the available diagnostic tests and immunization efficiency.

Overall, in the course of viral evolution, mutations occur randomly and those giving some key advantages to the virus are more likely to be selected. In particular, for SARS-CoV-2, several mutations prevalent in successful variants are located within the Spike protein. Different studies provide evidence that these mutations enable SARS-CoV-2 to evade the immune system more efficiently, even in individuals previously infected or completely immunized. In addition, the VOC’s Spike proteins also incorporate changes that increase SARS-CoV-2′s infectiveness and transmission efficiency.

The fast spread of many SARS-CoV-2 VOCs is a consequence of many factors, encompassing both host behavior and virus properties. While some Spike mutations lead to greater efficacy of infection by increasing the binding affinity to human ACE2 receptors, the same or a few additional mutations may expand the time that Spike remains bound to hACE2, shift the Spike population to the most favorable conformation for receptor binding and increase the number of functional Spike proteins per virion. Most importantly, direct evaluation of the reduced sensitivity of convalescent sera and the observation that vaccinated individuals, while asymptomatic or with mild symptoms, are becoming ever more important for the spread of VOCs, suggests the selection of mutations that allow the virus to evade the immune system is key to explaining the success of new variants and is probably going to continue as long as the numbers of novel infections remain high. Taken together these factors play a pivotal role in the emergence of fast-spreading SARS-CoV-2 variants of concern, including the Omicron variant.

## Figures and Tables

**Figure 2 viruses-14-00827-f002:**
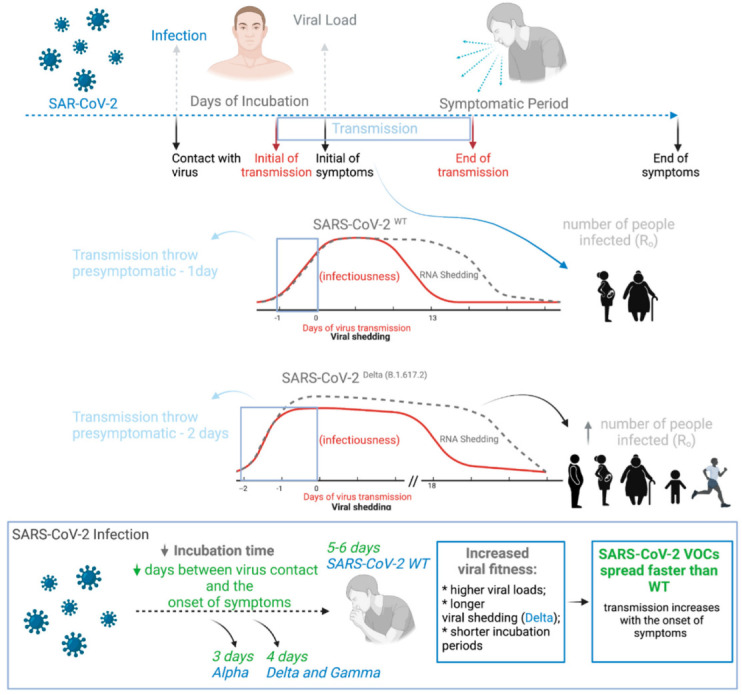
The kinetics of SARS-CoV-2 infection. The SARS-CoV-2 infection involves contact with the virus, viral incubation (days of presymptomatic viral replication), period of viral transmission (viral shedding, shown in the red line in the graphs), which is associated with increased viral loads in the infected person, the symptomatic period, from symptoms onset to recovery and the window for detection of viral RNA by diagnostic tests (viral RNA shedding, gray dashed lines in the graphs).

**Figure 3 viruses-14-00827-f003:**
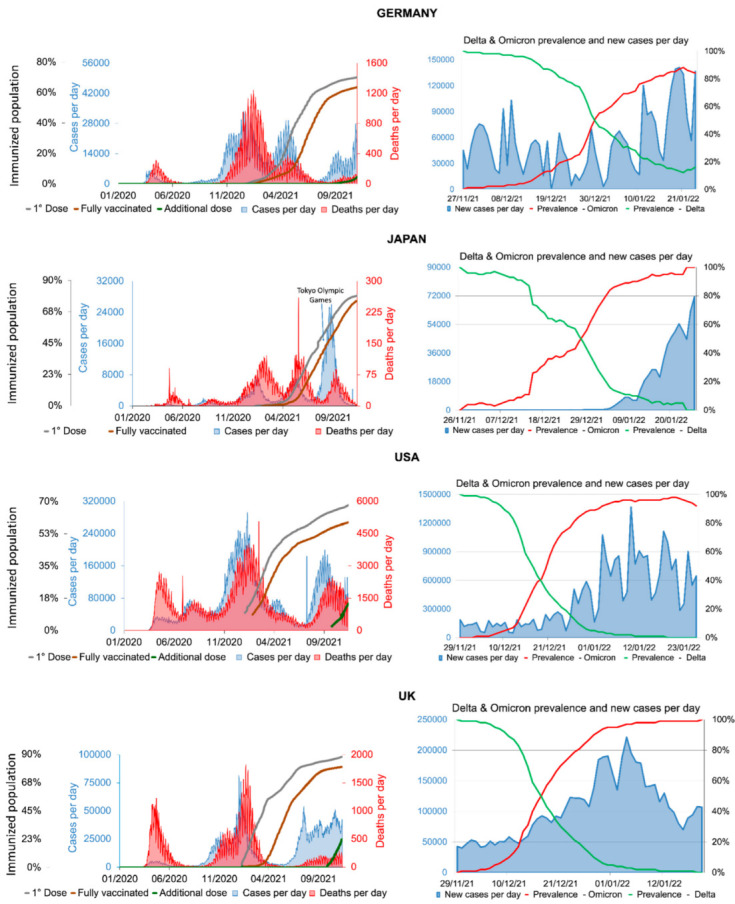
Number of COVID-19 cases and death rates, the effect of vaccination rate and prevalence of SARS-CoV-2 Delta and Omicron in the UK, Japan, USA and Germany. The (**left**) side shows the number of cases (red line), the number of deaths (blue line), the rate of vaccinated individuals with the first dose (gray line), second dose (fully vaccinated, brown line) and additional dose (green line). The (**right**) side shows the graph of the number of cases (blue line) and the percentage of the Delta variant (green line) and Omicron variant (red line) as a function of time. All data were obtained from World in Data, WHO, CDC, UK Gov and Outbreak [1,148,149,150,151].

**Figure 4 viruses-14-00827-f004:**
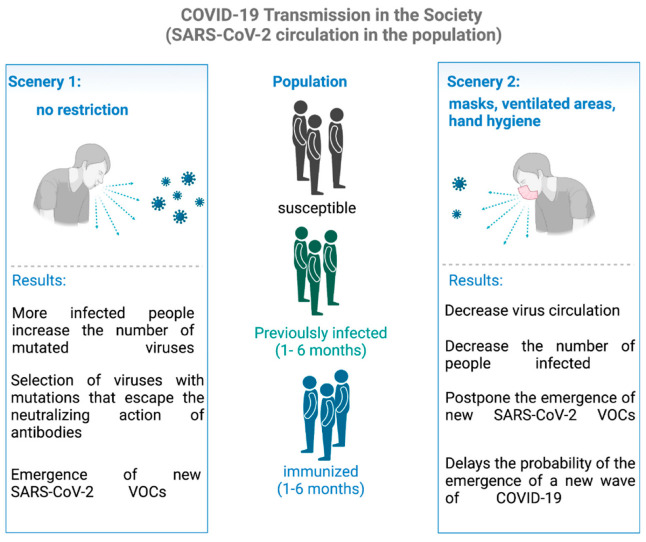
Hypothesis to explain the emergence of novel SARS-CoV-2 variants. Currently, the world population is composed of three groups: those who do not have neutralizing antibodies to fight infection by the virus (susceptible) and those who acquired it either because they were infected by the virus or because they had taken the vaccine. In scenario 1, in which society does not make any kind of restriction to prevent the spread of the virus, the emergence of VOC should occur faster than in scenario 2, in which society maintains measures to reduce the virus spread. These two scenarios are independent of population immunization, since the virus mutates its genetic information randomly, and selection for variants with more successful infection should prevail in the population. This natural selection seems to be directed towards variants that are able to evade the immune system and spread the virus faster. The SARS-CoV-2 neutralizing antibodies of completely immunized and previously infected individuals kept high until 6 months after immunization or virus infection [152].

**Table 1 viruses-14-00827-t001:** Kinetic parameters of RBD-hACE2 complex formation. Equilibrium dissociation constants (*K*_D_) calculated for RBD and its variants in the complex with dimeric hACE2 protein. Experimental *K*_D_ values were also measured using a trimeric Spike protein and its variants for interacting with hACE2.

	Spike Trimeric			Spike RBD		
Strains	*k*_a_ (10^5^ M^−1^s^−1^)	*k*_d_ (10^−4^ s^−1^)	*K*_D_ (nM)	*k*_a_ (10^5^ M^−1^s^−1^)	*k*_d_ (10^−4^ s^−1^)	*K*_D_^RBD^ (nM)
SARS-CoV^WT^	1.4 [6]	7.1 [6]	5.0 [6]	1.4–15.8 [51]	93–338 [51]	1.46–185 [51,52,53]
SARS-CoV-2^WT^	1.4 [6]	1.6 [6]	1.2 [6]	9.0 [54]	91.6 [54]	1.1–112.1 [45,52,53,55,56,57,58,59]
SARS-CoV-2^D614G^	1.6 [60]	1.7 [60]	1.0 [60]	ND	ND	0.38–12.8 [61]
SARS-CoV-2^Alpha^	0.1 [54]	1.7 [54]	1.6 [54]	13.0 [54]	15.5 [54]	0.5–57.1 [45,54,56,57,58,59,61]
SARS-CoV-2^Beta^	0.3 [54]	3.0 [54]	1.1 [54]	12.0 [54]	39.4 [54]	3.3–25.5 [45,54,56,61]
SARS-CoV-2^Gamma^	0.2 [54]	3.0 [54]	1.8 [54]	13.0 [54]	28.8 [54]	2.2 [54]
SARS-CoV-2^Delta^	ND	ND	ND	0.1 [62]	46.0 [62]	2.7–176 [22,62]

ND = not determined, *K*_D_ = *k*_d_/*k*_a._

**Table 3 viruses-14-00827-t003:** Sensitivity of sera from convalescent patients and individuals vaccinated with Moderna, Pfizer-BioNTech and Oxford-AstraZeneca vaccines. The sensitivity of neutralizing antibodies produced by convalescent patients or induced by vaccines is higher in ancestral SARS-CoV-2 and, in general, the performance of this sensitivity is worse against any SARS-CoV-2 variants.

Strains	Convalescent Patient (ID_50_)	Moderna (IC_50_/ID_50_)	Pfizer-BioNTech (IC_50_/FRNT_50_)	Oxford-AstraZeneca (FRNT_50_)
SARS-CoV^WT^	1500–8000 [122]	ND/ND	ND/ND	ND
SARS-CoV-2^WT^	1402 [91]	ND/3067 [91]	ND/1105 [123]	306 [123]
SARS-CoV-2^D614G^	1485 [91]	833 [124]/2906 [91]	695 [124]/ND	ND
SARS-CoV-2^Alpha^	1290 [91]	722 [124]/1578 [91]	626 [124]/337 [123]	131 [123]
SARS-CoV-2^Beta^	309 [91]	182 [124]/477 [91]	114 [124]/146 [123]	34 [123]

ND = not determined. IC_50_ and ID_50_ = titers in serum for neutralizing 50% SARS-CoV-2 in vitro. FRNT_50_ = focus reduction neutralization test 50 (μg/mL); titers in serum for neutralizing 50% SARS-CoV-2 in vitro.

**Table 4 viruses-14-00827-t004:** Sensitivity of sera induced by Sputnik V, Janssen, CoronaVac and Covaxin vaccines. The sensitivity of neutralizing antibodies induced by vaccines is superior in ancestral SARS-CoV-2 and, in general, the performance of this sensitivity is reduced against any SARS-CoV-2 variants.

Strains	Sputnik V IC_50_	Janssen IC_50_	CoronaVac ID_50_	Covaxin PRNT_50_
SARS-CoV^WT^	ND	ND	ND	ND
SARS-CoV-2^WT^	ND	ND	774.48 [125]	ND
SARS-CoV-2^D614G^	49.4 [126]	221 [124]/246 [127]	ND	ND
SARS-CoV-2^Alpha^	87.1 [126]	232 [124]/266 [127]	44.64 [125]	ND
SARS-CoV-2^Beta^	7.9 [126]	33 [124]/68 [127]	35.03 [125]	61.6 [128]
SARS-CoV-2^Gamma^	ND	72 [127]	ND	ND
SARS-CoV-2^Delta^	ND	30 [124]/154 [127]	24.5 [125]	69 [128]

ND = not determined. IC_50_ and ID_50_ = titers in serum for neutralizing 50% SARS-CoV-2 in vitro. PRNT50 = Plaque reduction neutralization test. Titers in serum for neutralizing 50% SARS-CoV-2 in vitro.

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
