# Peer review of "Severe Acute Respiratory Syndrome Coronavirus 2 Variants of Concern: A Perspective for Emerging More Transmissible and Vaccine-Resistant Strains"

_viruses, 2022, doi:10.3390/v14040827_

Round 1
Reviewer 1 Report
Anacleto Silva de Souza et al have described the article Severe Acute Respiratory Syndrome Coronavirus 2 Variants of 2 Concern: a perspective for emerging more transmissible and vaccine-resistant strains. The article has discussed the impact of mutation on VOCs transmissibility and use of COVID-19 practices and its impact. The study mainly focused on Spike gene mutation and its impact on VOC and then transmission and other factor like neutralization.
It would be nice to incorporate role of other genes and mutations of virus and is impact on breakthrough and re-infection in VOCs infected individuals .
Author Response
REVIEWER 1
It would be nice to incorporate the role of other genes and mutations of virus and is impact on breakthrough and re-infection in VOCs infected individuals .
Answer:
Thanks for you comments. Your question is very relevant, but unfortunately current knowledge on the effect of mutations in other proteins of the virus is much more limited than the extensive studies on the impact of mutations in the spike protein. It has recently been shown that mutations in the structural protein nucleocapsid, mainly in the linker region of the protein, increase mRNA packaging and delivery resulting in increased virus replication in in vitro assay using lung epithelial cells (DOI: science.org/doi/10.1126/science.abl6184). Therefore, different mutations present in VOCs may contribute in different ways to increase virus fitness, infectivity, transmission, severity of the COVID-19 infection (such as the D614G in the Spike protein and the P4715L in ORF1ab (DOI: https://doi.org/10.1016/j.micpath.2021.104831), and may also facilitate human SARS-CoV-2 re-infection.
The following sentence was added in the page 19 of the manuscript: "In addition to the effect of the mutations described in the Spike protein other mutations found in other SARS-CoV-2 proteins may contribute in different ways to increase virus fitness, infectivity, transmission, severity of the COVID-19 infection (such as the P4715L in ORF1ab [140]), and may also facilitate human SARS-CoV-2 re-infection. Interestingly, some mutations in the structural protein nucleocapsid increase mRNA packaging and delivery, RNA content per virion, resulting in increased virus replication [141]."
Reviewer 2 Report
Overall, this work is timely, an excellent review. This is an excellent paper, which, in my view, should be published.
Minor comments:
1) Figure3 dose not help to improve the review article. Author may need to delete the Figure3. And I do not understand what unclivaged/clivaged form is. Is this a prefusion/postfusion form?
2) Figure S3 B), Multiple sequence alignment should be prepared correctly.
Author Response
1) Figure3 does not help to improve the review article. Author may need to delete the Figure3. And I do not understand what unclivaged/clivaged form is. Is this a prefusion/postfusion form?
Answer: The spike cleaved protein refers to the spike protein that is naturally cleaved in the mature viral particle. Mutations in the Spike protein result in greater protection from the proteolysis of the spike protein resulting in an increased amount of complete Spike protein (uncleaved) on the virion surface. Figure 3 was moved to supporting information and the following sentence added in the figure legend: "In the topic 4 the spike cleaved protein refers to spike protein that is naturally cleaved in the mature viral particle. Mutations in the Spike protein result in greater protection from the proteolysis of the spike protein resulting in an increased amount of complete Spike protein (uncleaved) on the virion surface."
The word "clivaged" were replaced to cleaved in the Figure.
2) Figure S3 B), Multiple sequence alignment should be prepared correctly.
Answer: Thanks for the comment and we corrected the text formation of the multiple sequence alignment.